# Diagnostic Accuracy and Performance Analysis of a Scanner-Integrated Artificial Intelligence Model for the Detection of Intracranial Hemorrhages in a Traumatology Emergency Department

**DOI:** 10.3390/bioengineering10121362

**Published:** 2023-11-28

**Authors:** Jonas Kiefer, Markus Kopp, Theresa Ruettinger, Rafael Heiss, Wolfgang Wuest, Patrick Amarteifio, Armin Stroebel, Michael Uder, Matthias Stefan May

**Affiliations:** 1Department of Radiology, University Hospital Erlangen, Friedrich-Alexander-Universität (FAU) Erlangen-Nürnberg, Maximiliansplatz 3, 91054 Erlangen, Germany; jonas.kiefer@fau.de (J.K.); theresa.ruettinger@uk-erlangen.de (T.R.); rafael.heiss@uk-erlangen.de (R.H.); michael.uder@uk-erlangen.de (M.U.); 2Imaging Science Institute, Ulmenweg 18, 91054 Erlangen, Germany; patrick.amarteifio@siemens-healthineers.com; 3Martha-Maria Hospital Nuernberg, Stadenstraße 58, 90491 Nuernberg, Germany; wolfgang.wuest@martha-maria.de; 4Siemens Healthcare GmbH, Allee am Röthelheimpark 3, 91052 Erlangen, Germany; 5Center for Clinical Studies CCS, University Hospital Erlangen, Friedrich-Alexander-Universität (FAU) Erlangen-Nürnberg, Krankenhausstraße 12, 91054 Erlangen, Germany; armin.stroebel@uk-erlangen.de

**Keywords:** artificial intelligence, brain hemorrhage, automated detection, emergency department

## Abstract

Intracranial hemorrhages require an immediate diagnosis to optimize patient management and outcomes, and CT is the modality of choice in the emergency setting. We aimed to evaluate the performance of the first scanner-integrated artificial intelligence algorithm to detect brain hemorrhages in a routine clinical setting. This retrospective study includes 435 consecutive non-contrast head CT scans. Automatic brain hemorrhage detection was calculated as a separate reconstruction job in all cases. The radiological report (RR) was always conducted by a radiology resident and finalized by a senior radiologist. Additionally, a team of two radiologists reviewed the datasets retrospectively, taking additional information like the clinical record, course, and final diagnosis into account. This consensus reading served as a reference. Statistics were carried out for diagnostic accuracy. Brain hemorrhage detection was executed successfully in 432/435 (99%) of patient cases. The AI algorithm and reference standard were consistent in 392 (90.7%) cases. One false-negative case was identified within the 52 positive cases. However, 39 positive detections turned out to be false positives. The diagnostic performance was calculated as a sensitivity of 98.1%, specificity of 89.7%, positive predictive value of 56.7%, and negative predictive value (NPV) of 99.7%. The execution of scanner-integrated AI detection of brain hemorrhages is feasible and robust. The diagnostic accuracy has a high specificity and a very high negative predictive value and sensitivity. However, many false-positive findings resulted in a relatively moderate positive predictive value.

## 1. Introduction

Hemorrhagic stroke and traumatic brain injuries present with different types of intracranial hemorrhage (ICH). The lesions can be subtle and with low contrast to the surrounding tissues in non-contrast Computed Tomography (CT) of the head [1]. ICH may result in a rapid increase in intracranial pressure, in which case, early recognition and treatment can significantly reduce patient morbidity and mortality [2]. Fast and precise detection is crucial for further, possibly surgical, treatment [3]. The four-eyes principle is highly recommended in this scenario, but simultaneously personnel-intensive and sometimes unavailable. In addition, the experience level of the radiologists may be low in some situations, for example, during 24 h and night shifts, and the human diagnostic performance can decrease under fatigue [4]. Moreover, radiologists face an ever-increasing clinical workload, given the constantly growing volume of multimodality images [5]. Head CT is a highly cost-effective and widely available imaging technique that provides quick results. Additionally, with the help of new reconstruction algorithms, it is now possible to acquire images with lower doses of radiation, making it a more feasible option for patients [6]. The rapidly growing artificial intelligence (AI) applications have the potential to improve radiologists’ productivity and time management [7,8,9,10]. However, the implementation in clinical practice is still limited today [11]. Explainability and ethical considerations are vital features of a trustworthy AI. Seamless integration in the reporting process and evidence for a benefit on patient care is needed for acceptance. So far, most available applications require time-consuming and error-prone data transfers to AI servers for further processing and subsequent storage of the results in the archives. Most of these algorithms were well-evaluated pre-clinically in selected patient collectives [8,12]. However, the generalizability of these models to broader patient populations and the clinical prevalence of ICH are limited. Recently, a detection model for ICH was presented with full integration to the examination workflow of the scanner’s user interface. An additional clinical decision tree support system allows for the automatic application of the algorithm in the scenario of traumatic brain injury and stroke. The calculation has direct access to the raw data of CT head examinations without contrast injection. A binary result image can immediately be displayed after calculation on the scanner and archived to the PACS. This allows for a rapid and comprehensive assessment of the most time-critical findings in the trauma setting. However, it remains to be determined what test statistics can be achieved in daily clinical practice. This study aimed to test this scanner-integrated model for ICH detection in the clinical routine setting of a maximum care trauma department, following the null hypothesis that the algorithm cannot indicate ICHs with a high likelihood.

## 2. Materials and Methods

The local ethics committee (Ethics commission of the Friedrich-Alexander-University Erlangen-Nürnberg (ethikkommission@fau.de)) approved this monocentric retrospective study (231_21bc). All CT examinations included in the study were clinically indicated.

### 2.1. Study Population

In this single-center study, we evaluated all consecutive patients who underwent an unenhanced head CT examination in our emergency department from 23 February 2021 to 27 July 2021 for inclusion. The sample size was chosen to achieve the desired precision of the estimators of the sensitivity and the positive predictive value (PPV). These estimators are binary variables. Confidence intervals were estimated for binary variables with the software R Version 4.2.1 with function binominal test. For a sample size of 50 patients and an expected PPV of 50%, the half-length of the confidence interval is 14.2%. Other values of expected PPV yield smaller lengths of the confidence intervals (for the same sample size). For a sample size of 100 patients, the half length of the confidence interval is 10.2%. These lengths are deemed acceptable by the researchers. Patients eligible for inclusion were consecutive adults (≥18 years) with a clinical emergency indication for CT of the head, especially neurological deficits and trauma to the head. Pediatric patients and patients without AI-based results were excluded (Figure 1). A maximum of three follow-up examinations were included. The investigation resulted in n = 435 brain CT datasets for the indicated period and was then reviewed retrospectively by two independent readers (1 and 11 years of experience).

### 2.2. CT Technique

All examinations were performed on a single-source CT system (SOMATOM X.ceed, Siemens Healthcare GmbH, Forchheim, Germany) in the vicinity of the shock room and emergency department of a maximum care university hospital. The acquisition parameters are given in Table 1. We designed a manual user interaction for the investigator at the beginning of the examination that distinguishes traumatic head injuries and stroke patients. Trauma patients are scanned using a spiral scan. The brain is automatically scanned for hemorrhage using the AI we studied. Additionally, specific unfolding images are used to reconstruct the anatomical landmarks of the skull and brain surface. Stroke patients without preceding trauma undergo a sequential examination protocol, automatic analysis of the Alberta Stroke Program Early CT Score (ASPECT), and AI detection of ICH. This decision selection speeds up imaging and avoids errors when manually entering the CT protocol.

The sequential scanning technique is time-consuming and susceptible to eventual motion artifacts, and the options for post-processing are limited. However, the overall image quality, especially the corticomedullary differentiation in the ischemic stroke, is significantly better than in a spiral acquisition protocol [13,14].

Spiral scanning has a shorter scan duration and steady data acquisition, while exposing the patient to a lower mean radiation dose [13,15]. Moreover, Straten et al. defined the best image quality for brain tissue near the skull, where most traumatic intracranial hemorrhages occur [16].

All patients were preferably positioned in the head-positioning support, and the body-positioning support served as an alternative in case of incompliance with low head positions. Tilting (inclination of the head in the sagittal plane), torsion (lateral inclination of the head in the coronal plane), and rotation (turning of the head around the longitudinal axis) of the patient’s neurocranium were measured retrospectively in the case of positive findings to evaluate a potential influence of head positioning on the model’s performance (Figure 2).

Images in both techniques were reconstructed with a slice thickness of 5 mm and interval of 5 mm, leading to approximately 30 images in axial orientation for each case. Datasets with pronounced artifacts (motion or beam hardening) were identified and excluded for subgroup analysis. Radiation dose parameters were assessed as CT dose index (CTDI) and dose-length-product (DLP) from the examination protocol.

### 2.3. Clinical Report

Intracranial hemorrhage includes four or five main types of bleeding: (1) epidural hemorrhage (between the skull bone and the outermost membrane layer, the dura mater), (2) subdural hemorrhage (between the dura mater and the arachnoid membrane), (3) subarachnoid hemorrhage (between the arachnoid membrane and the pia mater), and intracerebral hemorrhage ((4) intraparenchymal and (5) intraventricular hemorrhage). Intra-axial presence of blood due to any other etiology, such as hemorrhagic contusion, hemorrhagic tumor, or infarct with hemorrhagic transformation, was also included in the definition of intracranial hemorrhage. Visual image analysis was performed as part of the gold standard reading using dedicated software (syngo.via VB60A, Siemens Healthcare GmbH, Erlangen, Germany). All datasets from the study population (n = 435) were evaluated by a radiology resident and a senior radiologist within the clinical routine without AI support. The preliminary report was immediately produced for rapid communication, and the final report was sent to the clinical information system after a thorough review by the second reader within 24 h. Additionally, a team of two radiologists, with 14 and 9 years of experience in trauma CT, reviewed these datasets retrospectively with an offset of at least one month, taking additional information like the clinical record, course, and final diagnosis into account. This combined consensus reading was finally used as the reference or gold standard for this study. Patients with preexisting brain defects were identified and excluded for subgroup analysis.

### 2.4. Automatic Brain Hemorrhage Analysis

“Brain Hemorrhage”, a CE-labeled Deep-Learning-Algorithm (Siemens Healthcare GmbH, Forchheim, Germany), analyzed all studies on the CT console as a separate reconstruction job. It can automatically identify suspicious datasets suggestive of possible intracranial hemorrhage [17]. The results are calculated without further interaction and are read-only (Figure 3). The following image requirements must be met for the calculation of results: CT scan of the head without contrast enhancement, image field containing the whole brain, coverage from Vortex to Crista Galli and External Occipital Protuberance, minimum length of 120 mm, Axial reconstruction, Matrix size 512 × 512, Layer thickness 4.0 mm and Layer increment 4.0 mm. The AI algorithm consists of a pre-processing and a detection stage. Input to the algorithm is non-contrast soft-kernel head CT reconstructions. During preprocessing, the brain orientation is normalized using anatomical landmark detection. The referenced algorithm is based on multi-scale deep reinforcement learning [18]. Brain extraction and exclusion of strong features (e.g., skull) are obtained by an image-to-image convolutional network trained with deep supervision and adversarial perturbations [19]. After this pre-processing, the case-level presence/absence of intracranial hemorrhage is detected using a set of deep, dense neural networks (details available in [20]) that extract features from axial and coronal orientations using DenseUNet subnetworks with segmentation-based deep supervision, combined with a classification head [20]. The network is trained end-to-end with voxel-level supervision on the hemorrhage mask (if available) and label supervision on global absence/presence. More than 28,000 volumes from patients above 18 years of age and without signs of surgical intervention in the images have been used to train the algorithm [17]. All results from our clinical cases were automatically stored in a hidden research PACS and, therefore, unavailable for the radiologists and the referring physicians. The results of the Brain Hemorrhage analysis were retrospectively classified as true positive, false positive, true negative, and false negative compared to the reference standard.

### 2.5. Statistical Analysis

Concordance, F1 score, sensitivity, specificity, positive predictive value (PPV), and negative predictive value (NPV) calculations were all performed in Microsoft Excel (Excel 365, Microsoft, Redmond, DC, USA). Patient positioning analysis was performed using the SPSS software for Mac, version 28 (IBM, Armonk, NY, USA), with a *p*-value of 0.05 as the threshold for statistical significance.

Descriptive statistics included mean, upper, and lower limits of the 95 percent confidence interval; median, variance, minimum and maximum values (range); and standard deviation (SD). We used the Mann–Whitney U-Test to evaluate differences between patient positioning in patients with true-positive (n = 51) versus false-positive (n = 39) results for ICH provided by the algorithm.

## 3. Results

### 3.1. Patients

We excluded three head CTs because no AI result images were found in the archives. All other results were successfully calculated and archived. There were 392 patients and 40 follow-up examinations among the 432 remaining cases. The median age of patients was 68.7 years. A total of 173 (43.8%) patients were female, and 222 (56.2%) were male. Traumatic fall was the most frequent reason for referral (n = 260; 59.8%), with the domestic fall of elderly patients dominating. Other reasons included polytraumas (n = 43; 9.9%), follow-up examinations (n = 40; 9.3%), traffic accidents (n = 13; 3.0%), consciousness disorders (n = 11; 2.5%), and inadequate wake-up response after surgery (n = 10; 2.3%). The remaining indications (n = 55; 17.7%) varied from epileptic seizures to inadequate wake-up reactions after intubation anesthesia to sudden onset of severe headaches.

### 3.2. CT Technique

The trauma protocol was assigned to 352 patients, and the stroke protocol to 80 patients. A flowchart of the study design and subgroup exclusions is shown in Figure 1. The mean tilting in the subgroup of patients with a positive AI detection of ICH was 6.7° ± 7.4°, rotation was 6.3° ± 6.0°, and torsion was 3.8° ± 3.5°. Exposure parameters were significantly different between both protocols (all *p* < 0.05). Detailed results are presented in Table 1. The overall mean CTDI was 44.1 mGy, and the mean DLP was 755.3 mGy × cm per patient. The radiation dose of the stroke protocol was significantly higher than in the trauma protocol (both *p* < 0.01).

### 3.3. Clinical Reports

The radiologists identified n = 52 (12.0%) patients with an intracranial hemorrhage. Eleven of these were follow-up examinations. The corrected ICH prevalence in our study collective was 10.4%.

The distribution of the different bleeding types was epidural hematoma (n = 1; 2%), subdural hematoma (n = 35; 67%), subarachnoid hemorrhage (n = 24; 46%), intracerebral hemorrhage (n = 30; 58%), and combined hemorrhages (n = 21; 40%). Hemorrhage volume averaged 4.8 cm ± 4.3 cm × 1.4 ± 1.3 cm (in the axial reconstruction plane) across all bleeding types.

The retrospective expert consensus (ex-post) reading overruled four initially negative reports (ex-ante), where the radiologists overlooked or misinterpreted very small hemorrhagic lesions and post-hemorrhagic findings. Another head CT was performed in two of these four initially negative cases, two six days later. The radiology report and the brain hemorrhage algorithm were positive for intracranial hemorrhage in these two follow-up CT examinations (Figure 4).

### 3.4. Automatic Brain Hemorrhage Analysis

Overall, the AI algorithm and reference standard (the combined consensus reading, Section 2.3) were consistent in 392 out of 432 cases (accuracy = 90.7%). Only one false-negative case was identified within the 52 positive cases. However, 39 positive detections turned out to be false positives. The diagnostic performance was calculated as sensitivity 98.1% (95% confidence interval: CI95, 94.3–100%), specificity 89.7% (CI95, 86.7–92.8%), positive predictive value (PPV) 56.7% (CI95, 46.4–66.9%), and negative predictive value (NPV) 99.7% (CI95, 99.1%–100%). The F1 score for the correct classification of an ICH was 71.8%.

Concordant results were found in 321 (91.2%) cases of the subgroup with the trauma protocol. Sensitivity was at 100% (CI95, 100%), specificity at 90.2% (CI95, 87.0–93.5%), PPV at 53% (CI95, 41.0–65.1%), and NPV at 100% (CI95, 100%). In the subgroup with the stroke protocol, 71 (88.8%) cases had the same results. Sensitivity was 94.1% (CI95, 82.9–100%), specificity 87.3% (CI95, 79.1–95.5%), positive predictive value (PPV) 66.7% (CI95, 47.8–85.5%), and negative predictive value (NPV) 98.2% (CI95, 94.7–100%). Overall descriptive statistics are shown in a cross table for better understanding (Table 2).

The single false-negative case was a tiny lesion of 4 × 2 mm (Figure 5). The false-positive rate was rather high (43% of the positive AI results). Therefore, we carried out some subgroup analyses of these cases. First, we evaluated the positioning of the patients’ heads in the gantry as a potential bias: tilting, torsion, and rotation. No statistically significant differences were found between the true positives and false positives either in tilting (*p* = 0.121), torsion (*p* = 0.309), or rotation (*p* = 0.541). Second, there were four patients with pronounced beam hardening (10%) and seven with motion artifacts (18%) in the subgroup of false positives, which was comparable to the rate in the true positives (8% and 16%). The corrected positive predictive value after excluding these cases was 58.2%. Third, we excluded all patients with chronic brain defects, ten from the true positives and eleven from the false positives. The adjusted PPV was 59.4%.

## 4. Discussion

Automatic brain hemorrhage detection via the inline AI algorithm of a CT system can immediately provide results with high accuracy (90.2%). The sensitivity, specificity, and NPV were at least 90%. The F1 score, more suitable for imbalanced distributions like in this study with a prevalence of only 12%, was moderate (71.8%). This limited performance for positive findings is mainly due to the high rate of false-positive findings, with a resulting PPV of only 56.7%. No relevant differences were found for the subgroups with limited image quality due to artifacts, chronic defects of the brain parenchyma, and poor positioning.

We retrospectively compiled a monocentric, consecutive database with a total of 435 CT scans of adult patients over slightly more than five months in the surgical department of our university hospital. No preselection was made regarding the ICH subtypes. So, the final data set was unbalanced, with a distribution of the cases depending on the clinical indication for head CT. Our radiology reports and consensus reading found a prevalence of 10.4% for ICH, slightly higher than other published prevalences (8.5%) in posttraumatic CT scans of the head [21]. Most of the examinations (81.5%) were performed with the trauma protocol, most likely because of the study's installation in the vicinity of the emergency room of a surgical department. The radiation dose of the spiral acquisition technique for trauma patients was significantly lower (−8%) than the sequential approach for stroke patients. This agrees with the publication of Pace et al., who reported a reduction of 25% in the spiral technique compared to the sequential protocol [13]. There was no significant difference in the algorithm’s performance in the two protocols. The accuracy was 91.2% in the trauma collective and 88.8% in the clinical suspicion of stroke scenario. In general, it appears reasonable that the relatively high false-positive rate may be explained by disturbing hyperdensities like artifacts, tumors, and defects, especially near the skull base [22]. For example, Kundisch et al. described a relevant detection rate for these findings in an algorithm from a different vendor [23]. Our first two subgroup analyses suggest that neither motion and beam hardening artifacts nor chronic brain defects are a reason for the low PPV, since the adjusted rate of false-positive findings was just a tiny bit lower (PPV: 63.0% compared to 56.7%). Also, in our third subgroup, no statistically significant differences were found in positioning the patients’ heads in the gantry for the true- and false-positive cases. Therefore, the presented model for ICH detection does not seem appropriate for unsupervised application to real-world clinical data, which is also clearly stated by the vendor [24]. The high number of false positives could have a substantial negative impact on the treatment of patients, ranging from extended hospitalization rates to fatal scenarios of unjustified surgical procedures. Also, the high number of false positives could increase the radiologists’ workload and personnel costs in the supervised scenario if the performance statistics are not well-known to the reading physicians.

Our dataset’s average patient age is 67.8 years, so a relatively high prevalence of intracranial calcifications may be present. For example, Saade et al. described the occurrence of calcified hyperdensities in up to 20% of elderly patients [25]. Smaller calcifications in 5 mm slices could produce false-positive findings that cannot be differentiated from small ICHs due to partial volume effects. These pseudo-hemorrhages could also contribute to the binary model’s high number of false-positive detections.

We compared these descriptive statistics with other previously published evaluations of algorithms from different vendors to further explore the performance’s validity. In 2018, Chilamkurthy et al. [26] described a deep-learning algorithm for detecting ICH, midline shifts, mass effects, and calvarial fractures in non-contrast head CT that reached an overall sensitivity for detecting ICH of 92.0%, which is slightly lower than our results. Another more recent AI algorithm analysis by Gruschwitz et al. [27], who reviewed around 900 cases, had a lower sensitivity of 91.4% and a specificity of 90.4%, similar to our results. However, their balanced patient collective was retrospectively selected with an approximately equal distribution of patients with and without hemorrhages. In contrast, Ojeda et al. [28] tested a novel convolutional network for ICH detection based on an extensive database of 7112 non-contrast head CT studies from two institutions in a cloud-based research scenario. Near-perfect results were reported with a specificity of 99.0%, sensitivity of 95.0%, and accuracy of 98.0% using a retrospectively collected validation dataset, compared to our prospective protocol selection and scanner-integrated calculation approach.

### Limitations

Our study has several limitations. First, we cannot provide information about the future effect on patient care using AI as a second- or even first-line reader. Our comparative study concept aimed for improved knowledge of its performance with actual clinical data. However, based on the very good test statistics, future studies providing the AI results to the reading radiologist appear reasonable.

Second, our defined endpoint of this study was a confidence interval of more than 90% for sensitivity and specificity. This resulted in only 52 positive ICH cases, and therefore a high risk for limited statistical power.

Third, there was a prevalence of ICH in our clinical routine collective (12%) compared to the training and validation datasets of the vendor (~50%). This could explain the substantial differences between the PPV in this study (56.7%) compared to the value in the product specification (94.1%) [24]. This underlines the need for standardization in AI market approval, clinically oriented AI training and validation, and prospective institutional evaluation in the respective populations.

Fourth, as Voter et al. also mentioned in their publication about a different ICH model, another potential limitation of our study is the assumption that the AI and the radiologists’ consensus detected the same findings in case of accordance [29]. Each may have identified separate, although concordant, results since the algorithm could not mark its detections. That may have artificially inflated the sensitivity of the model. Likewise, it is possible that both independently failed to identify the same ICH, and thus these cases were classified as true negative. Nevertheless, it seems implausible that enough ICHs were missed to alter our results significantly. Future design of the AI application for ICH detection could consider location and subtype to address these shortcomings.

Fifth, only binary result images were provided as results. No heatmap, segmentation, or annotation was provided to visualize the decision of the AI. However, explainability is a mandatory functional requirement to establish ethical AI in healthcare [30].

Sixth, we report the performance of this first DenseUNet that is FDA-approved, CE-labeled, and integrated into a CT scanner system. Other models may outperform in the preclinical setting but are as yet unavailable in the routine and still need to prove their real-world compatibility [10,31,32]. More evaluations with different kinds of AI models would be interesting in the future.

Finally, it remains unclear if a failure in reconstruction or archiving was the reason for the loss of three datasets in our study population (0.7%).

## 5. Conclusions

In conclusion, the new AI-based model for ICH detection in non-contrast head CT examinations presented promising results with very high sensitivity, negative predictive value, and reasonable specificity. However, many false-positive findings resulted in a low positive predictive value. Radiologists should be aware of the test statistics of AI models in clinical practice and remember that a positive test result does not always imply a hemorrhage. Still, a negative result rules out a hemorrhage with a very high probability.

## Figures and Tables

**Figure 1 bioengineering-10-01362-f001:**
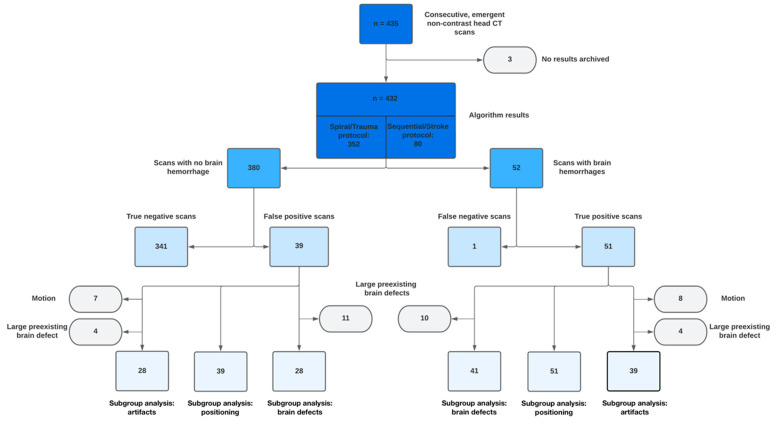
Summary of the predictions made by the artificial intelligence-based algorithm for the presence or absence of ICH. Three cases were excluded because the algorithm achieved no result. Subgroup evaluations of the true- and false-positive cases were performed to further investigate the positive predictive value.

**Figure 2 bioengineering-10-01362-f002:**
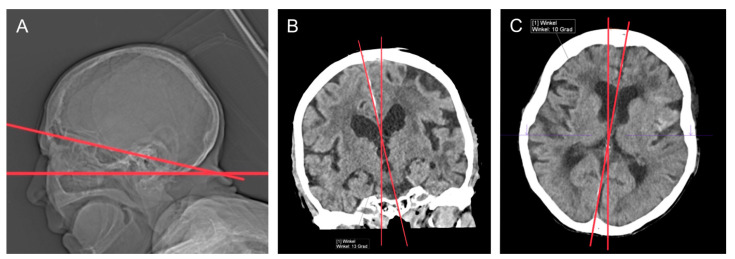
Examples of different positioning in the scanner. The red lines show the angles of deviation from the straight head position. The purple line is negligible. (**A**): tilting—inclination of the head in the sagittal plane; (**B**): torsion—lateral inclination of the head in the coronal plane; (**C**): rotation—turning of the head around the longitudinal axis.

**Figure 3 bioengineering-10-01362-f003:**
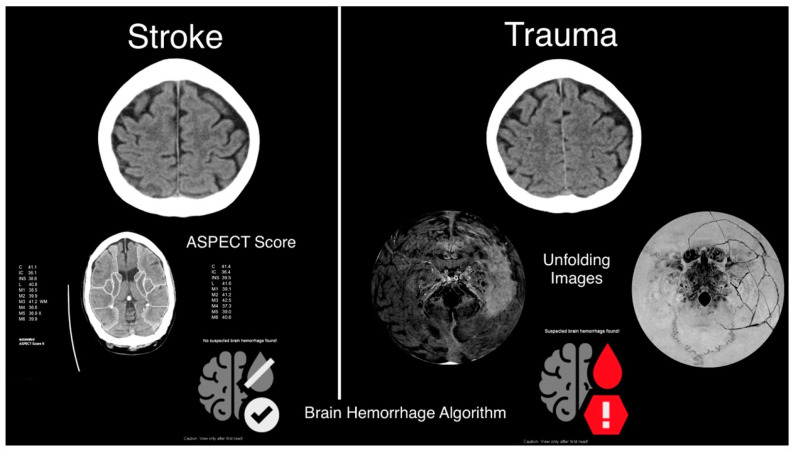
Presentation of the different examination protocol results. Both protocols show an example of a head CT slice, whereas the stroke side displays the calculation of the ASPECT Score and a negative result of the algorithm for bleeding. On the right side are the results of the brain and skull unfolding images, which show an intracranial hemorrhage with a skull fracture. The result of the algorithm is positive for bleeding on this side.

**Figure 4 bioengineering-10-01362-f004:**
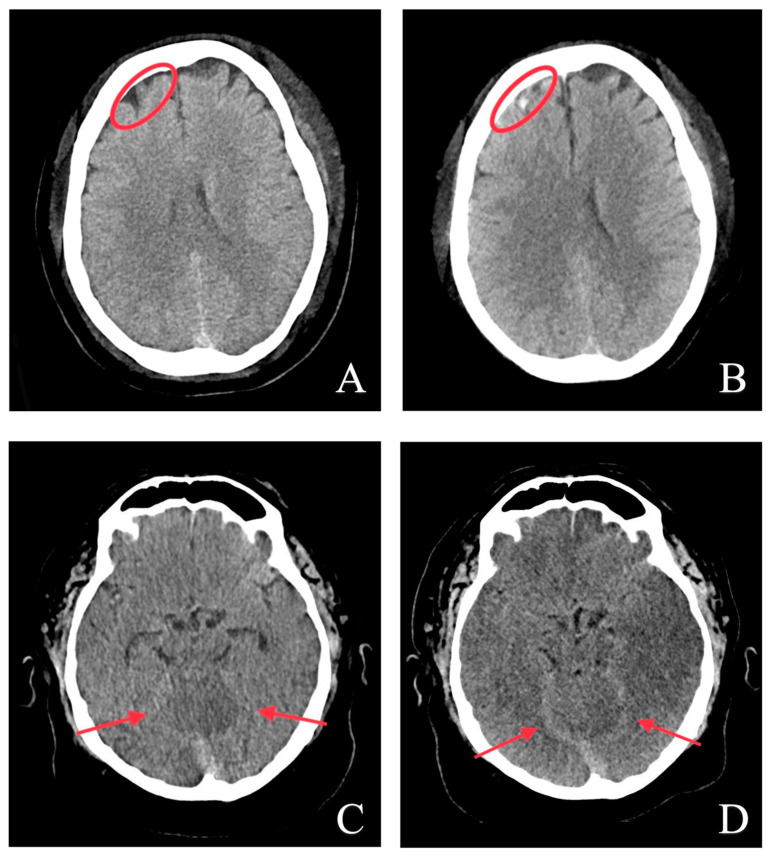
Representative images of the two cases where the initial radiology report was negative, but the AI algorithm was positive for intracranial hemorrhage (**A**,**C**). Due to the patient’s symptoms, the second images were taken a few hours (**B**) and some days later (**D**). The radiological report and algorithm results were positive in these follow-up studies ((**B**): frontal lobe and (**D**): tentorium cerebelli). The red circles and arrows show the location of the hemorrhages.

**Figure 5 bioengineering-10-01362-f005:**
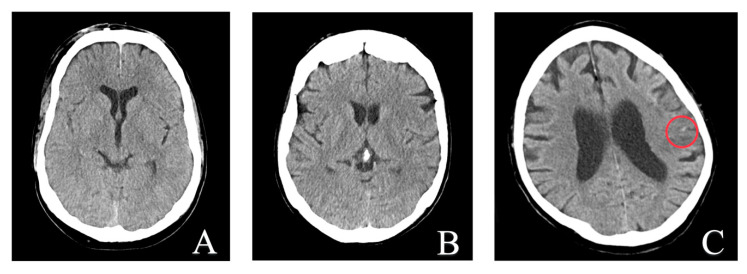
Representative images of false-positive and false-negative predictions of the AI model for intracranial hemorrhage (ICH) detection. (**A**,**B**) show non-suspicious scans that were incorrectly identified as ICH-positive. In both cases, we assume that calcifications were erroneously rated as ICH. (**C**) shows the single patient image where the algorithm missed the left temporal lobe hemorrhage. It was a very tiny lesion of 4 × 2 mm (red circle).

**Table 1 bioengineering-10-01362-t001:** Detailed summary of the standardized image protocol settings for non-contrast head CT with radiation parameters.

Protocol Selection	Stroke	Trauma
Collimation	128 × 0.6 mm	128 × 0.6 mm
Mode	Sequential	Spiral
Rotation time	0.5 s	0.5 s
Inline results	ASPECT score Brain Hemorrhage	Brain Hemorrhage Brain unfolding Skull unfolding
kV	120	120
IQ Level	282	282
Average scan length	16.5 cm	17.3 cm
CTDI	48.9 ± 6.6 mGy	43.0 ± 4.8 mGy
DLP	808.9 ± 146.7 mGy × cm	743.3 ± 107.8 mGy × cm

**Table 2 bioengineering-10-01362-t002:** Cross table of the AI performance for detection of intracranial hemorrhage in non-contrast enhanced CT of the head.

	Gold standard: Positive	Gold standard: Negative		
AI: Positive	True Positives: 51	False Positives: 39	90	PPV: 56.7%
AI: Negative	False Negatives: 1	True Negatives: 341	342	NPV: 99.7%
	52	380	432	
	Sensitivity: 98.1%	Specificity: 89.7%		

## Data Availability

The data presented in this study are available on request from the corresponding author. The data are not publicly available due to patients’ privacy.

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
