# Peer review of "Diagnostic Accuracy and Performance Analysis of a Scanner-Integrated Artificial Intelligence Model for the Detection of Intracranial Hemorrhages in a Traumatology Emergency Department"

_bioengineering, 2023, doi:10.3390/bioengineering10121362_

Round 1

Reviewer 1 Report

Comments and Suggestions for Authors

This retrospective study includes 435 consecutive non-contrast head CT scans. Automatic brain hemorrhage detection was calculated as a separate reconstruction job in all cases. The radiological report (RR) was always made by a radiology resident and finalized by a senior radiologist. A team of two radiologists additionally reviewed the datasets retrospectively, taking additional information like the clinical record, course, and final diagnosis into account. This consensus reading served as reference.

Statistics were carried out for diagnostic accuracy. Brain hemorrhage detection was executed successfully in 432/435 (99%) patient cases. The AI algorithm and reference standard were consistent in 392 (90.7%) cases. One false negative case was identified within the 52 positive cases. However, 39 positive detections turned out to be false positives. The diagnostic performance was calculated as a sensitivity of 98.1%, specificity of 89.7%, positive predictive value of 56.7%, and negative predictive value (NPV) of 99.7%.

(i)                Please explain “Full integration in the reporting process and evidence for a benefit on patient care is needed for acceptance. So far, most available applications require data transfers to AI servers for further processing and subsequent storage of the results in the archives. Most of these algorithms were well-evaluated pre-clinically in selected patient collectives

(ii)              The sentence is unclear. This interaction automatically adapts the acquisition and reconstruction protocol, potentially speeding up imaging and avoiding errors in CT protocol personalization. The differences in the protocols as selected by the user are: Trauma patients are examined with a spiral acquisition, the brain is automatically screened for hemorrhage using a dedicated AI, and anatomical orientations of the skull and brain surface are reconstructed by unfolding. Please rephrase it.

(iii)             Related papers. Voxelwise detection of cerebral microbleed in CADASIL patients by leaky rectified linear unit and early stopping. Seven-layer deep neural network based on sparse autoencoder for voxelwise detection of cerebral microbleed. Cerebral micro-bleeding identification based on a nine-layer convolutional neural network with stochastic pooling.

(iv)             How Spiral scanning has a shorter scan duration and steady data acquisition while exposing the patient to a lower mean radiation dose?

(v)              Motivation is unclear. Tilting (inclination of the head in the sagittal plane), torsion (lateral inclination of the head in the coronal plane), and rotation (turning of the head around the longitudinal axis) of the patient's neurocranium were measured retrospectively in the case of positive findings to evaluate a potential influence of head positioning on the model’s performance.

(vi)             In the conclusion part, it is equally important to reiterate and reinforce the motivation that was initially presented in the introduction. This serves to remind the readers of the main objectives of the study and provides a concise summary of the reasons why the research was conducted. Additionally, the conclusion can also briefly highlight the potential implications, practical applications, or future directions resulting from the study's findings, further emphasizing the importance of the research and its contribution to the field.

Author Response

Dear Reviewer,

Thank you for your comments and recommendations for improvement. We have done our best to implement your suggestions in our manuscript and to answer your questions below. 
We have marked our changes based on your suggestions in the manuscript in red. 

(i) We studied the first scanner-integrated AI algorithm for detecting intracranial hemorrhage. So, in contrast to other machine learning systems, there is no need to send the data to another platform to analyze it, and so there is no time delay in the result and no need for storage for another computer system. We wanted to test the algorithm's performance on an unbalanced patient collective. Other studies had balanced patient collectives with an almost equal number of patients with and without intracranial hemorrhage. Our study represents the real-life patient collective in an emergency room of a university hospital. 

(ii) We rephrased the sentences and highlighted them in red in the manuscript.

(iii) We introduced some new references in the introduction and highlighted them in orange. 

(iv) We would kindly refer you to our reference 7 for this question: 

Pace I, Zarb F. A comparison of sequential and spiral scanning techniques in brain CT. Radiol Technol. 2015;86(4):373-8.

(v) We performed this subgroup analysis to find reasons for the high number of false positive cases. We expected a poor positioning of the head might influence the outcome of the algorithm's decision, but surprisingly, it didn't. 

(vi) In the conclusion section, we wanted to highlight the algorithm's potential implications and practical application. Radiologists should know the model's performance statistics in clinical practice and consider what a positive or negative AI result implies. 

Reviewer 2 Report

Comments and Suggestions for Authors

See the attached file for minor comments

The choice of the protocol should be better explained for more clarity

In the introduction, a section on the role of brain CT would be beneficial

Comments on the Quality of English Language

Use the same verbal tense

Author Response

Dear Reviewer,

Thank you for your comments and recommendations for improvement. We have done our best to implement your suggestions in our manuscript and to answer your questions. We have marked our changes based on your recommendations in the manuscript with orange. 

Reviewer 3 Report

Comments and Suggestions for Authors

The authors evaluated a machine protocol to diagnose cerebral hemorrage in 435 patients where 51 had findings (some on repeat testing). Sensitivity, specificity and PVNeg parameters are excellent whereas PVpos value is about 56%,, which is explainable and often necessary. So the study evaluated the metod as being very good in my opinion.

A few mistakes: line 199 the torsion parameter is missed. Figure 1 the middle line of boxes with no 30 and 1 have strange interpretations of positive and negative, respectively.

Author Response

Dear Reviewer,

Thank you for your comments and recommendations for improvement. We have done our best to implement your suggestions in our manuscript.
We have marked our changes based on your suggestions in the manuscript in purple.

Reviewer 4 Report

Comments and Suggestions for Authors

The paper is well written and has a potential for publication. I have nonetheless, as a data scientist,  two issues that need to be solved:

-1 there is a quite relevant number of false positives. What would happen to these subjects if the system were left alone in taking a decision? What is the human cost of such a decision?

-2 it is not said which is the AI algorithm that  detects the hemorragy. many ML algorithm exist. Maybe some of them would perform better that the one used in this system. Not only but we could also have DL approaches that could give even better and more accurate results.

Synthetically, as a scientist, Ia cannot trust completely in a system which is presented as a black box whose content is neither described nor its algorithms compared with alternatives (for a example a list of possible alternatives in terms of ML algorithms can be found here: AA. VV. Is a COVID-19 second wave possible in Emilia-Romagna (Italy)? Forecasting a future outbreak with particulate pollution and machine learning. Computation 8(3), 2220 doi: 10.3390/computation8030074

Comments on the Quality of English Language

None

Author Response

Dear Reviewer,

Thank you for your comments and recommendations for improvement. We have done our best to implement your suggestions in our manuscript and to answer your questions below. 

1) Radiologists should always look at the CT head images without checking the AI's results first. The algorithm should only support the decision-making of finding an intracranial hemorrhage and never be left alone.

2) In section 2.4. we mentioned the specific algorithm "Brain hemorrhage", a CE-labeled Deep-Learning-Algorithm by Siemens Healthcare GmbH, Forchheim, Germany.

Round 2

Reviewer 1 Report

Comments and Suggestions for Authors

The paper lacks comparison with state-of-the-art approaches. particularly some methods from top journals, such as IEEE or ACM Trans.

Author Response

Many thanks for the suggestion to enhance our manuscript with further literature. We included the recommended articles from review round 1 in the introduction and added a section about methodological limitations in the discussion section. Articles from the recommended journals were used as references. Unfortunately, no direct comparisons of statistical values seem possible, because these methodological articles generally use different measures (e.g. dice score, location accuracy, intersection over union, ...) compared to our clinically oriented evaluation.

Reviewer 4 Report

Comments and Suggestions for Authors

The paper has almost not changed after my comments; I wonder how much is useful such a review process if the authors do not intend ry to modify their paper in response to the reviewers' comments. A few lines in the response letter is just non sensical.I do not approve.

Comments on the Quality of English Language

no comment here

Author Response

Many thanks for your revision. We are deeply sorry that our prior comments did not apply to your expectations. Therefore, we extended the description and discussion of the mentioned topics (supervision and comparison of AI methods) in the manuscript to enhance its scientific content.

We added an entirely new paragraph in the discussion section about the critically high number of false positive cases and about the appropriateness or inappropriateness of such models for unsupervised application on clinical patient data. We also discussed the potential human consequences and costs. (highlighted in blue)

We added an additional paragraph  about different technical approaches in AI for image evaluation in the limitations section (#6) of the discussion and referenced your suggested paper about different kinds of AI models there and in the introduction. (highlighted in red)

We also added specific description of the AI model (DenseUNet) and added another reference where detailed specifications could be found (highlighted in blue). We hope the changes will clarify the working methods and background of the algorithm in a better way. 

Round 3

Reviewer 1 Report

Comments and Suggestions for Authors

Accept

Reviewer 4 Report

Comments and Suggestions for Authors

Acceptable quality